# Antioxidant Properties of Green Coffee Extract

**Anna Masek [1],\*** , **Malgorzata Latos-Brozio [1]** , **Joanna Kałużna-Czaplińska [2]** ,
**Angelina Rosiak [2]** **and Ewa Chrzescijanska [2]**

[1] Faculty of Chemistry, Institute of Polymer and Dye Technology, Lodz University of Technology, Stefanowskiego 12/16, 90-924 Lodz, Poland; malgorzata.latos-brozio@dokt.p.lodz.pl

[2] Faculty of Chemistry, Institute of General and Ecological Chemistry, Lodz University of Technology, Zeromskiego 116, 90-924 Lodz, Poland; joanna.kaluzna-czaplinska@p.lodz.pl (J.K.-C.); angelina.rosiak@p.lodz.pl (A.R.); ewa.chrzescijanska@p.lodz.pl (E.C.)

\* Correspondence: anna.masek@p.lodz.pl

**Abstract:** An infusion of green coffee is a commonly consumed beverage, famous for its health-promoting properties. Green coffee owes its properties to the richness of active phytochemicals. The aim of this study was to determine the components of green coffee bean extracts and their properties. The scope of research included gas chromatography-mass spectrometry (GC-MS), Fourier transform infrared spectroscopy (FTIR) and Ultraviolet-Visible spectroscopy (UV-Vis) spectroscopy; the electrochemical determination of the behavior of green coffee extract; and the determination of antioxidant properties by colorimetric spectroscopic methods (ABTS, DPPH, FRAP and CUPRAC). Water and ethanol extracts from green coffee were characterized by significant antioxidant properties and a high capacity to reduce transition metal ions. Voltammetric tests showed that the solution has good antioxidant properties in view of it contains many polyphenolic compounds that oxidize in the potential range tested.

**Keywords:** green coffee; electrochemical oxidation; antioxidant activity

## 1. Introduction

The purpose of this research was to analyze the antioxidant activity of the substances present in green coffee. Coffee is a very common plant consumed by many societies. Moreover, beans of coffee are rich source of phytocompounds with noted high antioxidant activity. The phenolic compounds can be found in plant materials, and they have been testified to have biological effects, containing high antioxidant and antibacterial activity.

The main phenolic ingredients identified in green coffee beans are chlorogenic and caffeic acids, which have antimutagenic, anticancer, antibacterial and antioxidative effects. In beans of green coffee, most phenols are linked to sugars as glycosides. Phenolic acids such as chlorogenic, ferulic and caffeic acid are identified in the form of an ester bound to the cell wall, creating highly complex polysaccharide structures. Phenolics in their conjugated form limit their bioavailability due to their high molecular mass and hydrophilicity [1,2].

Polyphenolic compounds could be correlated with their antioxidant activities. To analyze the capacity of antioxidants to reduce free radicals, methods consisting of single electron transfer chemical reactions (SET) were utilized. These procedures are considered by ease of solution preparation, stability of reaction mixtures, repeatability and relatively easy determination measures. Due to the mentioned benefits, these methods are widely utilized to determine the antioxidant activity of natural extracts. SET tests include two analogous procedures, ABTS and DPPH. The ABTS technique permits the investigate of hydrophilic and hydrophobic antioxidants, while the DPPH procedure permits the determination of only hydrophobic antioxidants. Furthermore, the reduction ability of natural

antioxidants is tested as the ability to reduce metals. The procedures FRAP and CUPRAC are based on the reduction of transition metal ions—iron and copper [3].

An alternative to traditional methods for determining the antioxidant activity of different extracts or plant compounds is electrochemical procedures, which in last years have attracted interest from scientists [4–6]. The benefit of electrochemical techniques is that they enable rapid, easy and inexpensive determinations that, in some cases, permit measurements in the presence of colored or masking compounds that can interfere with measurements by other techniques, e.g., spectrophotometric. Further benefits of electrochemical techniques is that it is possible to examine experimental parameters such as peak potential ($E_{pa}$) and peak current ($i_{pa}$), which are important in analyzing the antioxidant activity of the extracts tested. Low values of oxidation potential ($E_{pa}$) reflect the trend of a given molecule to donate electrons, and thus to show its good antioxidant activity. Cyclic voltammetry (CV) is the most usually utilized method to study the properties of electrode processes, giving data on the thermodynamics of electrode reactions and electron transfer kinetics, as well as coupled chemical reactions or adsorption processes. Other electrochemical technique applied in determining the antioxidant activity of the extracts is differential pulse voltammetry (DPV), which is typify by good detection limits and high resolution. The use of this method makes it possible to eliminate adsorbing compounds because, in this technique, they are not electroactive and, therefore, there are no visible peaks on the voltamperogram [7–14].

The aim of this study was to determine the antioxidant properties of compounds found in coffee extract by various methods, including electrochemical, using cyclic voltammetry (CV) and differential pulse voltammetry (DPV). A further objective of the work was to establish the correlation between antioxidant activities deduced from cyclic voltammograms and those previously determined by utilizing established spectrophotometric methods. Moreover, a ranking of coffee was gained, utilizing the antioxidant composite index (AC), a parameter that assigns equal weight to all the antioxidant activity assesses. Antioxidant activities applying ABTS, DPPH, FRAP and CUPRAC tests for determination of total phenolic content were performed by UV-visible spectrophotometry, FTIR spectrophotometry and gas chromatography-mass spectrometry (GC-MS). This manuscript describes different in vitro tests that characterize and compare the antioxidant activity and mechanism of action of green coffee and its bioactive substances. The study revealed a general decrease of radical scavenging capacity related to native substances of plant origin. The scientific novelty of the work is a combination of different techniques for determining the activity of green coffee extracts. So far, no such comprehensive research on green coffee extracts has been done. What is more, spectrophotometric determinations summarize the results for extracts obtained in ethanol and in water, received at different times of brewing coffee. The available literature lacks assessment of the connection among the type of solvent and antioxidant activity and reduction of transition metal ions. This publication supplements the deficiencies in the literature.

## 2. Materials and Methods

### 2.1. Reagents and Chemicals

Chemical reagents applied for testing were as follows: 2,2′-azino-di-(3-ethylbenzthiazoline sulfonic acid) (ABTS, assay ≥98%, Sigma Aldrich, Saint Louis, MO, USA), potassium peroxodisulfate (99.99%, Sigma Aldrich, Saint Louis, MO, USA), (±)-6-Hydroxy-2,5,7,8-tetramethylchromane-2-carboxylic acid (Trolox, 97%, Sigma Aldrich, Saint Louis, MO, USA), 1,1-Diphenyl-2-picrylhydrazyl radical (DPPH, ≤100%, Sigma Aldrich, Darmstadt, Germany), 2,4,6-Tris (2-pyridyl)-s-triazine (TPTZ, ≥99.0% (HPLC), Sigma Aldrich, Buchs, Switzerland), ferric chloride $FeCl_3$ (pure P.A., Chempur, Piekary Slaskie, Poland), hydrogen chloride solution HCl (40 mM, Chempur, Piekary Slaskie, Poland), sodium acetate buffer solution (0.3 M, pH 3.6, Chempur, Piekary Slaskie, Poland), 2,9-Dimethyl-1,10-phenanthroline (Neocuproine, assay ≥98%, Sigma Aldrich, Beijing, China), cupric chloride $CuCl_2$ (0.01 M, Chempur, Piekary Slaskie, Poland), ammonium acetate ($NH_4Ac$) buffer solution (1.0 M, pH 7.0, Chempur,

Piekary Slaskie, Poland), ethanol (pure P.A., 96%, POCH, Gliwice, Poland) and methyl cyanide (pure P.A., 99.5%, POCH, Gliwice, Poland).

Methyl cyanide, 3-(3,4-Dihydroxycinnamoyl)quinic acid, 3,4-dihydroxybenzeneacrylic acid, (+)catechin, trans-4-hydroxycinnamic acid, 4-O-Caffeoylquinic acid, (−)epicatechin, ethyl acetate, trans-4-Hydroxy-3-methoxycinnamic acid, formic acid, 3,4,5-Trihydroxybenzoic acid, hexane, ellagic acid, 5-Hydroxy-1,4-naphthoquinone, 3,4′,5,7-Tetrahydroxyflavone, methyl alcohol, 3,3′,4′,5,5′,7-Hexahydroxyflavone, trans-5-O-Caffeoylquinic acid, N,O-bis(trimethylsilyl)trifluoroacetamide, 3,3′,4′,5,6-Pentahydroxyflavone, quercetin 3-glucoside, quercetin 3-rhamnoside, quercetin-3-rutinoside hydrate, 3,5-Dimethoxy-4-hydroxycinnamic acid, 3,5-Dimethoxy-4-hydroxybenzoic acid, trimethylchlorosilane, quercetin and (+)-catechin were purchased from Sigma-Aldrich (Steinheim, Germany). Ultrapure water was prepared in the laboratory, utilizing a SimplicityTM Water Purification System (Millipore, Marlborough, MA, USA).

### 2.2. Method of Preparation of Green Coffee Extracts for Antioxidant Analysis

Ethanol solution: Beans of green coffee were ground in a coffee mill (organic green coffee Honduras SHG, naturally grown, 100% Arabica.) The coffee came from a specialized farm in the Santa Barbara region, grown at an altitude of 1700 m above sea level, on volcanic hills with access to clean water from nearby rivers. Plant material was extracted using a 5-fold volume of 70% ethyl alcohol under continuous mixing conditions (250 RPM—revolutions per minute, 20 °C). The extraction was performed in the dark at 20 °C for 7 days. The extracts of ground green coffee were concentrated to constant mass by utilizing a rotary evaporator under reduced pressure conditions, at 30 °C.

Water solutions: Ground coffee beans were flooded with water at 90 °C and brewed under cover for 10 min or 30 min. Then, the plant material was filtered off on filter paper, and the solutions were concentrated to constant mass by utilizing a rotary evaporator under reduced pressure conditions, at 30 °C.

### 2.3. Measurement Methods

#### 2.3.1. Chromatographic Analysis

First, 5.0066 g of the powdered green coffee beans was extracted with hexane (150 mL, 4 h), in a Soxhlet apparatus (labmed, HK, Lodz, Poland), to isolate the lipid fraction. The obtained fraction was derivatized with a mixture of N,O-bis(trimethylsilyl)trifluoroacetamide and trimethylchlorosilane (100:1 *v/v*, 30 min, 75 °C) and analyzed by gas chromatograph–mass spectrometer (GC-MS, Agilent Technologies, Santa Clara, CA, USA). The solid extraction residue, after decaffeination (in ethyl acetate), was extracted with methanol in a Soxhlet apparatus (150 mL, 3 h). The obtained fraction was derivatized as described above and also analyzed by GC-MS. Analyses were carried out with an Agilent 6890N Gas Chromatograph equipped with a HP-5MS capillary column (30 m × 250 μm × 0.25 μm), coupled to Agilent 5973N Mass Selective quadrupole detector (Agilent Technologies, Santa Clara, CA, USA). The column oven was programmed for the temperature starting at 50 °C. Then, the temperature increased by 15 °C per minute, to reach 210 °C, and remained at this temperature for 1 min. Next, the temperature was increased from 210 to 230 °C, with a 5 °C change per minute, and then from 230 to 300 °C, by 15 °C change per minute. The last stage was to keep the column oven at the final temperature 300 °C for 3 min. Carrier gas (helium) flow was at a constant flow rate of 0.7 mL/min. The ion source and detector temperatures were 230 and 150 °C, respectively. The mass range was set between m/z 50 and 750.

The compounds were identified by utilizing the NIST14 and Wiley's spectra libraries. The study included compounds which had content over 0.1%. The compounds from the column (mainly siloxanes) were not included in the results.

### 2.3.2. FTIR (Fourier Transform Infrared) Spectroscopy and UV-Vis (Ultraviolet-Visible) Spectroscopy

The FTIR spectrum is a "fingerprint" showing the chemical compounds and their structure, which are present in extracts from green coffee. A Nicolet 670 spectrophotometer (Thermo Fisher Scientific, Waltham, MA, USA) was utilizing for the FTIR tests. Ground green coffee samples and also water or ethanolic extracts were putted at the infrared beam output. As the result of the determination, oscillating spectra were gained, the examination of which permits determination of the functional groups that the radiation interacted with.

The UV-Vis spectrum of green coffee extracts were made from a mixture of 0.2 mL of each extract and 1.8 mL of 70% ethyl alcohol or distilled water. The sample was examined at 190–1100 nm, utilizing a UV-spectrophotometer (Evolution 220, Thermo Fisher Scientific, Waltham, MA, USA) at 25 °C.

### 2.3.3. Cyclic and Differential Pulse Voltammetry

To assess the electrochemical oxidation mechanism and the kinetics for the flavones under investigation, cyclic voltammetry (CV) and differential pulse voltammetry (DPV) were used with an Autolab analytical unit (EcoChemie, Utrecht, Holland). The analyzer was controlled by using the Graduate Program in the Environmental Science (GPES) Program. A three-electrode system was used for the measurements consisting of a reference electrode, an auxiliary electrode (platinum wire) and a working electrode—platinum with geometric surface area of 0.5 cm$^2$. The potential of the working electrode was measured vs. a ferrocenium/ferrocene reference electrode (Fc$^+$/Fc) couple, as recommended by International Union of Pure and Applied Chemistry (IUPAC) [15,16]. The reference electrode was made of platinum wire immersed in a solution of ferrocene c = 1 × 10$^{-3}$ mol L$^{-1}$ in 0.1 mol L$^{-1}$ (C$_4$H$_9$)$_4$NClO$_4$ in acetonitrile placed in a glass tube with a very tiny hole (diameter w 0.2 mm) at the bottom. The tiny hole allowed electrochemical contact between the electrolyte in the reference electrode compartment and that in the reactor compartment. The next step was coulometric oxidation to obtain an equivalent of ferrocene ion concentration ferrocenium (Fc$^+$/Fc).

Determination of antioxidants was performed by using CV and DPV techniques. CV and DPV were recorded in the potential range from 0 to 1.9 or 2.1 V. CV was recorded with various scan rates (0.01 to 1 V s$^{-1}$). DPV was recorded in the same potential range, with modulation amplitude of 25 mV and pulse width of 50 ms (scan rate 0.01 V s$^{-1}$). Before the measurements, the solutions were purged with argon in order to remove dissolved oxygen. During measurements, an argon blanket was kept over the solutions. All experiments were carried out at room temperature [17].

### 2.3.4. Antioxidant Activity Tested Using ABTS and DPPH Procedure

The antioxidant capacity of green coffee extracts was tested by ABTS and DPPH procedures. Green coffee extracts concentrated to a constant weight (obtained according to "Method of Preparation of green coffee extracts for antioxidant analysis") were made into solutions with concentrations of 1–4 mg/mL. The weighed concentrated green coffee extracts were dissolved in the appropriate amount of ethyl alcohol (70%)—a solvent commonly used for ABTS and DPPH procedures.

These procedures are based on reduction of radicals 2,2'-azino-bis (3-ethylbenzothiazoline-6-sulphonic acid) (ABTS) and 2,2-diphenyl-1-picrylhydrazyl (DPPH).

The ABTS$^{\bullet+}$ radical was obtained by the reaction of a 6 mM ABTS solution in water with potassium persulfate (2.45 mM) without light at 25 °C for 16 h before use. The absorbance of the ABTS$^{\bullet+}$ dilution was regulated with ethanol to 0.70 ± 0.02 at 734 nm at 25 °C. In the next step, the diluted ABTS$^{\bullet+}$ dilution (4.0 mL) was mixed with a 40 μL aliquot of all examined solution (2 mg mL$^{-1}$) or Trolox in ethyl alcohol. The absorbance was measured at 734 nm after 2 min at 25 °C, using a UV-spectrophotometer.

The DPPH dilution in ethanol (2.0 mL) at a concentration of 40 mg mL$^{-1}$ (0.1 mM) was inserted to 0.5 mL of an 70% ethanol solution containing 0.02 mg mL$^{-1}$ of green coffee-DPPH solution, which has a purple color, with a maximum absorbance at 517 nm. The progress of the reaction

was spectrophotometrically monitored. Ethyl alcohol (70%) was used as a blank in both ABTS and DPPH methods.

The level of inhibition (%) of free radicals ABTS and DPPH was computed according to the following Equation (1):

$$\text{Level of inhibition (\%)} = [((A_{CS} - A_E)/A_{CS}) \times 100] \tag{1}$$

where $A_{CS}$ is the absorbance of the control sample without extracts, and $A_E$ is the absorbance in the presence of green coffee extract.

The level of inhibition (%) of absorbance was computed by utilizing the standard curve prepared with solution of Trolox (% inhibition level-μM Trolox). The result of green coffee extract on reduction $ABTS^{\bullet+}$ and DPPH is referred to as the Trolox equivalent antioxidant capacity (TEAC) [18].

### 2.3.5. Examination of Reduction of Transition Metal Ions by FRAP and CUPRAC Procedures

Green coffee extracts concentrated to a constant weight (obtained according to "Method of Preparation of green coffee extracts for antioxidant analysis") were made into solutions with concentrations of 1–4 mg/mL. The weighed concentrated green coffee extracts were dissolved in the appropriate amount of in pure water for FRAP and CUPRAC methods.

The ability of ethanolic and aqueous green coffee extracts to reduce the ferric ion ($Fe^{3+}$-TPTZ complex) under acidic conditions was carried out by utilizing ferric reducing antioxidant power assay (FRAP). The FRAP reagent was freshly prepared by mixing 25 mL of acetate buffer solution (0.3 M, pH 3.6), 2.25 mL of TPTZ dilution (10 mM TPTZ in 40 mM hydrogen chloride solution) and 2.25 mL of ferric chloride (20 mM) in a pure water solution. The reaction mixture was stirred and incubated at 37 °C for 20 min. In the next step, the increase in absorbance of the ferrous form with blue color ($Fe^{2+}$-TPTZ complex) was measured at 595 nm, using a UV-spectrophotometer.

The CUPRAC procedure is similar to the FRAP method and involves the reduction of $Cu^{2+}$ to $Cu^{1+}$. Approximately 0.25 mL (0.01 M) of $CuCl_2$ was mixed with 0.25 mL of an ethanol solution of neocuproine ($7.5 \times 10^{-3}$ M) and 0.25 mL of buffer solution, $CH_3COONH_4$ (1 M), in a test tube, followed by the addition of different concentrations of green coffee extract. The total volume of samples was increased to 2 mL with pure water. After 30 min of incubation at 25 °C, the absorbance at 450 nm was measured against a reagent blank (pure water).

The reducing power of ferric (FRAP) and cupric (CUPRAC) ions was computed as follows (Equation (2)):

$$\Delta A = A_1 - A_R \tag{2}$$

where $A_R$ is the absorbance of the reagent test, and $A_1$ is the absorbance after reaction [19,20].

### 2.4. Statistical Analysis

Computations were performed for the means and standard deviations of 3 independent extractions ($n = 3$). Statistical analysis was applied for the assessment of the means and then carried out by utilizing a Fischer LSD test (the significance level was set at $p < 0.05$).

## 3. Results and Discussion

### 3.1. Chromatographic Analysis of Green Coffee

The study began with an analysis of the chemical composition of green coffee beans. The results of the chromatographic analysis are shown below.

### 3.1.1. Organic Compounds in Hexane Extract of Beans of Green Coffee

The chromatogram of the hexane extract of green coffee beans is shown in Figure 1A. Table 1 shows the main organic compounds in the hexane extract of beans of green coffee. Significant amounts

of saturated (palmitic, stearic and arachidic) and unsaturated (palmitelaidic, oleic and 17-octadecynoic) fatty acids and acylglycerols (1-monolinolein, glycerol monostearate) were determined. Palmitic acid and linoleic acid are the two main components of the lipid fraction. Some of the compounds determined may be potentially related to the antioxidant properties of beans of green coffee. Linoleic acid belongs to the omega-6 fatty acid group, which can act as antioxidants [21,22]. The lycopene derivative and β-tocopherol, which are also natural antioxidants [23], were determined in the sample.

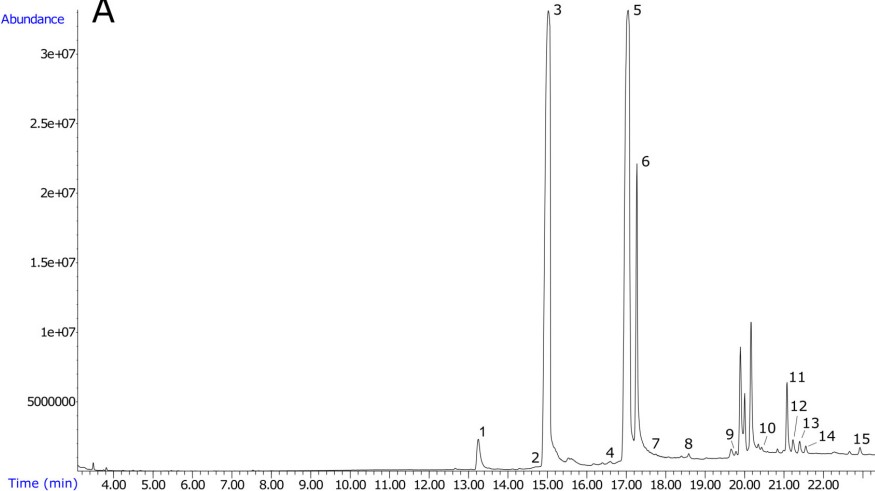

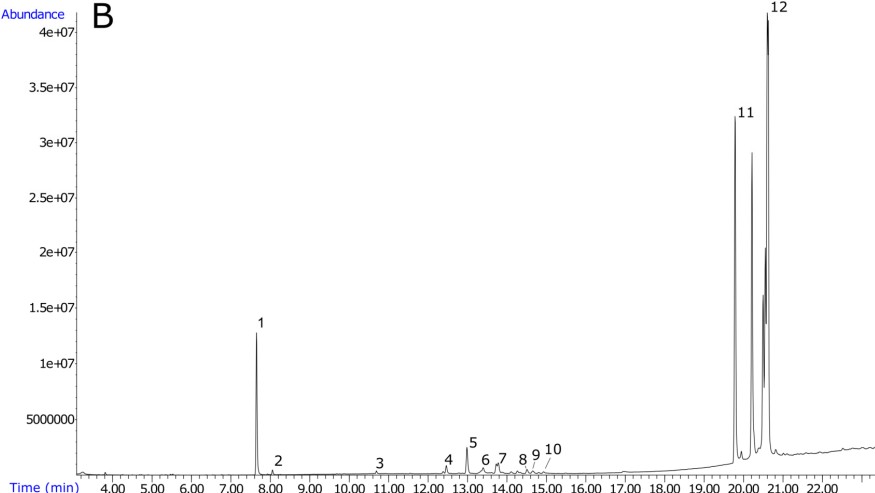

**Figure 1.** (**A**) The chromatogram of hexane extract of green coffee beans. The peak numbers in the figure correspond to the compounds in Table 1. (**B**) The chromatogram of methanol extract of green coffee beans. The peak numbers in the figure correspond to the compounds in Table 2.

**Table 1.** Main organic compounds determined in hexane extract of green coffee beans.

| Peak | Retention Time (min) | Area (%) | Plant Compound |
|---|---|---|---|
| 1 | 13.25 | 1.68 | Caffeine |
| 2 | 14.71 | 0.13 | Palmitelaidic acid, TMS derivative |
| 3 | 15.03 | 35.16 | Palmitic Acid, TMS derivative |
| 4 | 16.60 | 0.19 | Oleic Acid, (Z)-, TMS derivative |
| 5 | 17.05 | 32.37 | Linoleic acid, TMS |
| 6 | 17.27 | 9.96 | Stearic acid, TMS derivative |
| 7 | 17.61 | 0.47 | 17-Octadecynoic acid, TMS derivative |
| 8 | 18.59 | 0.37 | Arachidic acid, TMS derivative |
| 9 | 19.78 | 0.19 | 2-Oleoylglycerol, 2TMS derivative |
| 10 | 20.43 | 0.45 | Methyl glycocholate, 3TMS derivative |
| 11 | 21.08 | 1.88 | 1-Monolinolein, 2TMS derivative |
| 12 | 21.23 | 0.52 | Glycerol monostearate, 2TMS derivative |
| 13 | 21.40 | 0.45 | Lycopene, 1,1′,2,2′-tetrahydro-1,1′-dimethoxy-, all-trans- |
| 14 | 21.55 | 0.26 | Ethyl iso-allocholate |
| 15 | 22.93 | 0.21 | β-Tocopherol, TMS derivative |

### 3.1.2. Organic Compounds in Methanol Extract of Beans of Green Coffee

The chromatogram of the methanol extract of green coffee beans is represented in Figure 1B. There are twelve main compounds identified in the methanol extract (Table 2). Disaccharide–sucrose is the main component present in the sample and is the most abundant carbohydrate in green coffee beans [24]. Among the compounds in the methanol extract are sugar alcohols (polyols): glycerol, D-Mannitol, Myo-Inositol and D-Pinitol, and quinic acid, which is one of the major polyphenols in green coffee beans [25].

**Table 2.** Main organic compounds determined in methanol extract of green coffee beans.

| Peak | Retention Time (min) | Area (%) | Plant Compound |
|---|---|---|---|
| 1 | 7.65 | 5.03 | Glycerol, 3TMS derivative |
| 2 | 8.06 | 0.19 | Glycerol 1,2-diacetate |
| 3 | 10.69 | 0.20 | Triethanolamine, 3TMS derivative |
| 4 | 12.47 | 0.47 | D-Fructose, 5TMS derivative |
| 5 | 12.99 | 1.44 | Quinic acid |
| 6 | 13.40 | 0.75 | α-D-Glucopyranosiduronic acid, 3-(5-ethylhexahydro-2,4,6-trioxo-5-pyrimidinyl)-1,1-dimethylpropyl 2,3,4-tris-O-(trimethylsilyl)-, methyl ester |
| 7 | 13.78 | 0.66 | D-Mannitol, 6TMS derivative |
| 8 | 14.51 | 0.29 | Myo-Inositol, 6TMS derivative |
| 9 | 14.66 | 0.24 | D-Pinitol, pentakis(trimethylsilyl) ether |
| 10 | 14.93 | 0.19 | Palmitic Acid, TMS derivative |
| 11 | 19.95 | 0.93 | D-(+)-Turanose, octakis(trimethylsilyl) ether |
| 12 | 20.61 | 29.81 | Sucrose, 8TMS derivative |

### 3.2. FTIR (Fourier Transform Infrared) Spectroscopy and UV-Vis (Ultraviolet-Visible) Spectroscopy

Figures 2A and 1B show the spectra of green coffee and its extracts. Green coffee extracts are rich in polyphenolic compounds, especially phenolic acids, such as hydroxybenzoic acid derivatives (gallic acid and vanilla acid) and also hydroxycinnamic acid derivatives (ferulic acid, p-coumaric acid and caffeic acid). Moreover, these extracts contain chlorogenic acids, which are powerful antioxidants. Moreover, (+)-Catechin, a compound from the flavonoid group, was also found in green coffee extracts [26].

Figure 2A summarizes the FTIR spectrum of ground green coffee and the spectra of ethanol and water extracts (brewing time 10 min and 30 min). The green coffee spectrum shows functional groups characteristic of cellulose, hemicellulose and lignin (3680–2950 cm$^{-1}$-OH stretching; 2960–2860 cm$^{-1}$ - C-H stretching; 1470–1430 cm$^{-1}$-O-CH$_3$; 1090–970 cm$^{-1}$-C-O-C). Alkyl, aliphatic and aromatic compounds are visible in the range of 1730–1700 cm$^{-1}$, while 900–700 cm$^{-1}$ corresponds to C-H stretching from aromatic hydrogen compounds [27,28]. Reaction with ethyl alcohol and brewing coffee beans in water permits the extraction of active phenolic compounds from green coffee. A valuable group of plant compounds present in green coffee are polyphenols, including flavonoids. According to Heneczkowski et al. [29], the ranges 1612–1598 cm$^{-1}$, 1570–1560 cm$^{-1}$ and 1488–1452 cm$^{-1}$ are typical

for the aromatic ring vibration. Another specific bands corresponding to the phenol group vibrations are C-OH deformation vibrations (1370–1308 cm$^{-1}$), C-OH stretching vibrations (1172–1112 cm$^{-1}$) and, in the case of sulfonic derivatives, vibrations are related to those substituents in ranges 1188–1180 cm$^{-1}$ for SO$_2$ $\nu_{asym}$ and 1080–1024 cm$^{-1}$ for SO$_2$ $\nu_{sym}$ [29]. For ethanol and aqueous extracts, the ranges of wavenumber corresponding to active compounds often overlap with the ranges corresponding to pure solvents (3390 and 3360 cm$^{-1}$-OH groups, 2980 cm$^{-1}$-CH groups, and 1090 cm$^{-1}$ and 1050 cm$^{-1}$-C-O groups) [30].

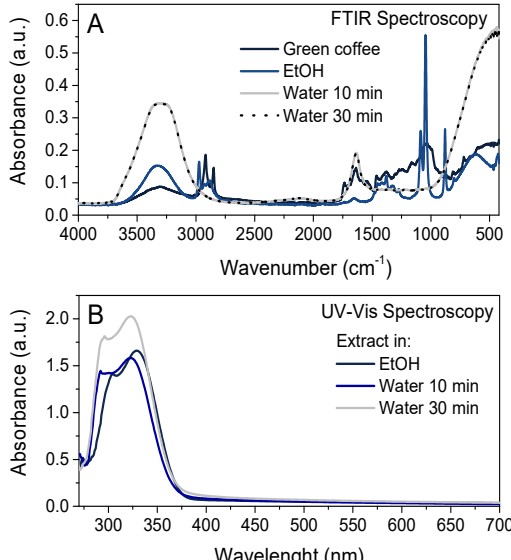

**Figure 2.** The FTIR spectra (**A**) and UV-Vis spectra (**B**) of extracts of green coffee.

Figure 2B shows the UV-Vis spectra of green coffee extracts. According to Song et al. and Rojas et al. [31,32], the UV-Vis spectra of phenolic acids have maxima of absorbance in the range of about 290–350 nm. Moreover, other active compounds, the flavones and related glycosides, illustration two strong absorption peaks at 240–280 nm and 300–380 nm [33]. Thus, the peaks with maxima at 337, 325, 322, 302, 296 and 288 nm on the UV-Vis spectra of green coffee ethanolic and water extracts characterize for mixture of active phenolic acids, flavones and their glycosides.

### 3.3. The Electrochemical Behavior of Green Coffee Extract

Voltammetric studies is used to assess the antiradicalproperties combined in various extracts of plant origin, as well as in beverages and biological systems, by many researchers [34], providing information on range and peak values. The dependence of the current on the potential of the indicator electrode was presented by the electrochemical reactions of the substances in solution on the tested electrodes. Electrochemical behavior of solutions of coffee nut extracts are contained in Figure 3. The cyclic voltammetry (CV) and differential pulse voltammetry (DPV) methods were used, because may be described by higher resolution.

On the voltamperogram for the coffee extract solution (Figure 3), four electro-oxygenation peaks are visible; the first peak (Peak I) is at 0.28 V potential, the second peak (Peak II) is at 0.96 V potential, the third poorly formed peak (Peak III) is at 1.23 V potential and the fourth peak (Peak IV) is at 1.49 V potential. In the reverse polarization cycle, one peak is visible at 0.52 V. Voltammetric tests show that the solution has good antioxidant properties because it contains many phytocompounds that oxidize in the potential field.

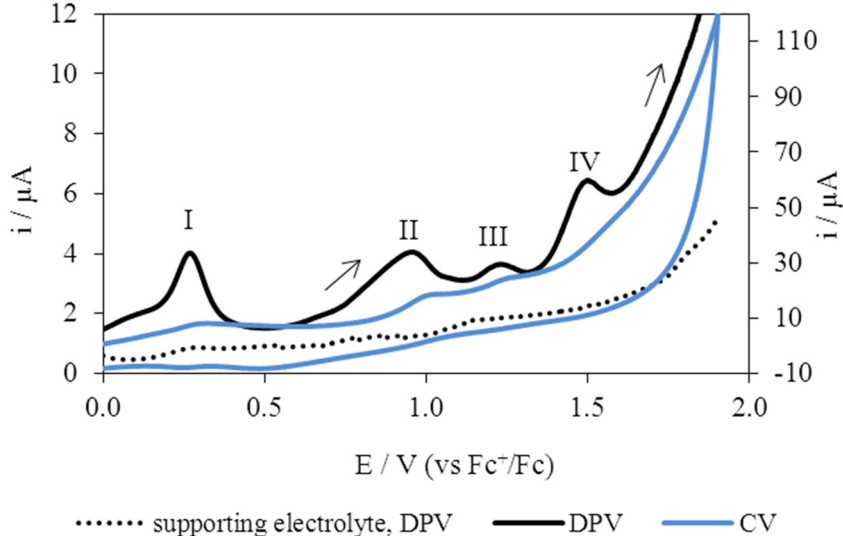

**Figure 3.** Cyclic voltammetry (CV) and differential pulse voltammetry (DPV) electrooxidation of extract of green coffee at Pt electrode; c = 20 mg/dm$^3$ in 0.1 M (C$_4$H$_9$)$_4$NClO$_4$ in acetonitrile; v = 0.05 V/s.

An electrochemical index (AC) was proposed, taking into account the main voltammetric parameters, peak potential (E$_{pa}$) and peak current (I$_{pa}$). Thus, the lower the potential (thermodynamic parameter), the higher the electron donor ability is, and the higher the peak current (kinetic parameter), the higher the amount of electroactive species is. The AC was calculated by using Equation (3):

$$AC = I_{pa1}/E_{pa1} + I_{pa2}/E_{pa2} + \ldots . + I_{pan}/E_{pan} \tag{3}$$

where I$_{pan}$ and E$_{pan}$ correspond to current and potential values for each anodic peak observed in the CV and DPV voltammograms.

The calculated values are presented in Table 3.

**Table 3.** Peak potentials (Ep) and currents (Ip) determined from cyclic voltammetry (CV), differential pulse voltammetry (DPV) and antioxidant capacity (AC).

| Method | Peak I | | Peak II | | Peak III | | Peak IV | | AC$_{total}$ |
|---|---|---|---|---|---|---|---|---|---|
| | E$_p$ (V) | i$_p$ (µA) | E$_p$ (V) | i$_p$ (µA) | E$_p$ (V) | i$_p$ (µA) | E$_p$ (V) | i$_p$ (µA) | |
| CV for v = 0.05 Vs$^{-1}$ | 0.29 | 7.95 | 1.01 | 18.22 | 1.25 | 24.49 | 1.56 | 44.68 | |
| DPV | 0.28 | 3.97 | 0.96 | 4.05 | 1.23 | 3.65 | 1.49 | 6.45 | |
| AC for CV | | 27.41 | | 18.03 | | 19.59 | | 28.64 | 93.68 |
| AC for DPV | | 14.17 | | 4.21 | | 2.96 | | 4.32 | 25.69 |

The AC is calculated by means of the peak current (I$_{pa}$) and peak potential (E$_{pa}$), in which the first parameter would be directly proportional to the antioxidant power, while the second would be inversely so. The AC provides a useful measure of the antioxidant power of products. Since the antioxidant power is usually correlated with phenolic compounds, such products may also be evaluated by means of total phenol content.

*3.4. Antioxidant Capacity of Green Coffe Extracts*

The next step of the study was the determination of the antioxidant capacity of green coffee extracts by using spectrophotometric methods (Figure 4 and Table 4). Figure 4 shows the activity of ethanolic green coffee extracts and aqueous extracts obtained as a result of brewing ground coffee beans for 10 and 30 min. Antioxidant activity is shown as the capacity to reduce (% inhibition) free radicals (ABTS and DPPH) (Figure 4A,B) and as the Trolox equivalent antioxidant capacity (TEAC) (Table 3). Green coffee extracts prepared in ethanol or water, with concentrations from 1 to 4 mg/mL, were tested,

and it was found that, with increasing concentration, the % inhibition of ABTS and DPPH increased. The ABTS inhibition (%) of 4 mg/mL extract of green coffee was 90.0 ± 0.20% for ethanol, 92.2 ± 0.19% for water at 10 min and 97.4 ± 0.25% for water at 30 min. Significant ability of green coffee extracts to reduce DPPH radicals was also found. The DPPH inhibition (%) of 4 mg/mL extract of green coffee was 63.9 ± 0.15% for ethanol, 69.6 ± 0.23% for water at 10 min and 81.6 ± 0.29% for water at 30 min. Analysis of antioxidant activity by ABTS and DPPH procedures showed that both the ethanol green coffee extract and aqueous solutions have good antioxidant properties. Another spectrophotometric analysis was the determination of the ability of green coffee extracts to reduce iron transition metal ions (FRAP method; Figure 4C) and copper (CUPRAC method; Figure 4D). The FRAP unit determines the capacity to reduce 1 mole of iron (III) to iron (II), and the CUPRAC unit to reduce 1 mole of copper (II) to copper (I). As in the case of ABTS and DPPH methods, it was observed that, with increasing concentration of green coffee extract, there were increases (in the range of 1–4 mg/mL) of their capacity to reduce iron and copper ions. All green coffee extracts exhibited good ability to reduce ions of transition metal; however, a slightly lower ability was found for aqueous extracts obtained by brewing coffee for 10 min. Strong antioxidant properties of green coffee extracts and the significant ability to reduce iron and copper metal ions should be combined with a high content of active compounds, such as polyphenolic compounds. The comparable activity of the ethanolic extract and aqueous solutions has proved that, thanks to the traditional brewing of green coffee, infusions with good antioxidant properties can be obtained.

All green coffee extracts were characterized by strong antioxidant activity; however, it was less than the capacity to reduce radicals ABTS and DPPH by green tea extract. Green tea extract, sold under the name Polyphenol 60, contained many polyphenolic compounds, such as gallic acid, procyanidin B1, (−)-epigallocatechin, (+)-catechin, (−)-epigallocatechingallate, (−)-epicatechin, (−)-epicatechingallate and flavonols. The extract obtained from green tea had very strong antioxidant properties: A solution with a concentration of 363 μg/mL exhibited the capacity to reduce ABTS free radicals, which was 93.6% and DPPH radicals 78.3% [18].

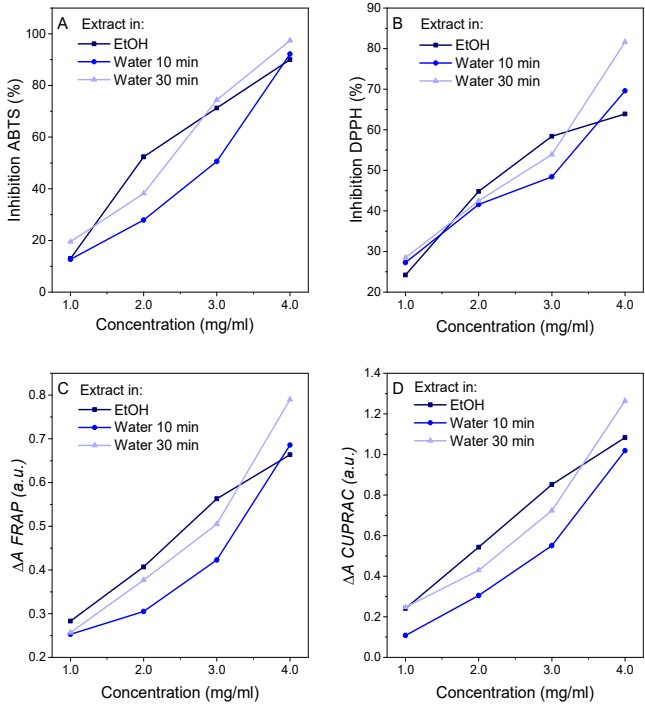

**Figure 4.** Antioxidant properties of green coffee extracts (ethanol and water prepared with 10 and 30 min of brewing the coffee) made by ABTS (**A**), DPPH (**B**), FRAP (**C**) and CUPRAC (**D**) techniques.

**Table 4.** Antioxidant capacity of green coffee extracts carried out ABTS and DPPH methods referred to as the Trolox equivalent antioxidant capacity (TEAC).

| Concentration of Green Coffee Extract (mg/mL) | EtOH | Water 10 min | Water 30 min |
|---|---|---|---|
| ABTS-TEAC (mmolT/100g) | | | |
| 1.00 | 46.6 ± 0.64 | 50.8 ± 0.54 | 73.1 ± 0.71 |
| 2.00 | 110.1 ± 0.94 | 55.2 ± 0.35 | 81.2 ± 0.36 |
| 3.00 | 94.1 ± 0.23 | 65.8 ± 0.15 | 105.5 ± 0.17 |
| 4.00 | 89.4 ± 0.17 | 91.2 ± 0.23 | 95.3 ± 0.23 |
| DPPH-TEAC (mmolT/100g) | | | |
| 1.00 | 78.5 ± 0.51 | 98.5 ± 0.42 | 96.5 ± 0.48 |
| 2.00 | 85.0 ± 0.26 | 74.2 ± 0.26 | 81.5 ± 0.31 |
| 3.00 | 67.7 ± 0.14 | 56.7 ± 0.29 | 68.9 ± 0.23 |
| 4.00 | 57.2 ± 0.11 | 61.8 ± 0.19 | 72.1 ± 0.22 |

## 4. Conclusions

Extensive analysis of the antioxidant properties of green coffee extract allowed for a thorough correlation of the test results by different methods. The objective of the research was extremely interesting and significant from the point of view of high consumption of this plant around the world. Studies have confirmed the strong dependence of the composition of the extract on the properties and activity of substances contained in green coffee beans.

Strong antioxidant properties of green coffee extracts and their significant ability to reduce iron and copper metal ions should be combined with a high content of active compounds, such as polyphenolic compounds. Moreover, the electroanalytical measurements and electrochemical index concept proved to be rapid to use and appropriate tools to evaluate the AC of extracts of coffee, and they can be applicable to other food sources rich in plant antioxidants with noted high antioxidant activity.

**Author Contributions:** Conceptualization, A.M. and M.L.-B.; methodology, A.M., E.C. and A.R.; software, M.L.-B.; validation, M.L.-B., E.C., J.K.-C. and A.R.; formal analysis, A.M.; investigation, A.M.; resources, A.M.; data curation, A.M.; writing—original draft preparation, A.M., M.L.-B. and E.C.; writing—review and editing, A.M.; visualization, M.L.-B.; supervision, A.M.; project administration, A.M.; funding acquisition, A.M. All authors have read and agreed to the published version of the manuscript.

**Funding:** This study was supported by the National Centre for Research and Development (NCBR) project: LIDER/32/0139/L-7/15/NCBR/2016.

**Conflicts of Interest:** The authors declare no conflict of interest.

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
