# Peer review of "Antioxidant Properties of Green Coffee Extract"

_forests, doi:10.3390/f11050557_

Round 1
Reviewer 1 Report
The aims of the article should be clearly stated at the end of the Introduction part.
In the introduction part the authors should include reliable data regarding chemical composition of green coffee beans especially polyphenolic compounds. Also a quick overview of usually methods applied regarding the antioxidant activity.
Figure 1 related to Table 1- in Table 1 there are 15 peaks in the chromatogram appear 8 main peaks and the rest are minor. I suggest to the authors to insert in the chromatogram the number corresponding to each peak.
How the authors identified Peak no 13 and 15? Lycopene and β-tocopherol? Using GC-MS?
“The lycopene derivative and β- tocopherol, which are also natural antioxidants [20], were determined in the sample.”- Using GS-MS? On what kind of column?
Line 290-291 Concentration in the range of 1 to 4 mg/ml of what? It is not clear? A polyphenolic extract? How the authors measured these concentrations?
What about statistical analysis?
The missing point of the article is comparing the data obtained regarding antioxidant activity. At least the classical methods (DPPH, FRAP) there are a lot of data’s.
These voltametric methods are used for the first time on coffee beans?
Author Response
Institute of Polymer and Dye Technology
Technical University of Lodz
90-924 Lodz, ul Stefanowskiego 12/16, Poland
Tel.: +48 42 631 32 23, Fax: +48 42 636 25 43
April 27, 2020
Forests, MDPI
Dear Editor,
We are resubmitting our revised paper entitled “Antioxidant properties of green coffee extract” by, Anna Masek, Malgorzata Latos-Brozio, Joanna Kałużna-Czaplińska, Angelina Rosiak, Ewa Chrzescijanska with a request to reconsider it for publication in Forests (MDPI).
We have carefully considered the Editor and Reviewers' comments. The manuscript was revised exactly according to these comments. The list of responses to the reviewer’s comments and corrections made in the manuscript is attached below.
The manuscript has not been previously published, is not currently submitted for review to any other journal, and will not be submitted elsewhere before a decision is made by this journal.
For correspondence please use the following information:
corresponding author: Anna Masek
Institute of Polymer and Dye Technology
Technical University of Lodz
90-924 Lodz, ul Stefanowskiego 12/16, Poland
Tel.: +48 42 631 32 93
Fax: +48 42 636 25 43
e-mail: anna.masek@p.lodz.pl
Yours sincerely,
PhD, DSc, Anna Masek
Associate Professor
Major Revision
Response to Reviewers:
|
Comments |
Changes by Authors |
|
Chromatographic analysis should be better described and discussed. The resolution of Figure 1 should be improved.
|
The text was corrected according to the reviewer comments. The results of the chromatographic analysis were extended and described in more details. The Figure 1 was attach as separate .tif file.
|
|
Comments by Reviewer #1 |
Changes by Authors |
|
The aims of the article should be clearly stated at the end of the Introduction part. |
It has been corrected. |
|
In the introduction part the authors should include reliable data regarding chemical composition of green coffee beans especially polyphenolic compounds. |
As suggested Reviewer, information has been added in the introduction part. |
|
Also a quick overview of usually methods applied regarding the antioxidant activity. |
A quick overview of commonly used methods for antioxidant activity has been added in the introduction part. |
|
Figure 1 related to Table 1- in Table 1 there are 15 peaks in the chromatogram appear 8 main peaks and the rest are minor. I suggest to the authors to insert in the chromatogram the number corresponding to each peak.
|
The Figure 1 was corrected according to the reviewer comments.
|
|
How the authors identified Peak no 13 and 15? Lycopene and β-tocopherol? Using GC-MS? “The lycopene derivative and β- tocopherol, which are also natural antioxidants [20], were determined in the sample.”- Using GS-MS? On what kind of column?
|
Compounds were identified by GC-MS analysis and also based on spectral libraries (NIST14 and Wiley’s spectra libraries) . Appropriate information was included in the description of sample preparation for chromatographic analysis (lines 119-135). |
|
Line 290-291 Concentration in the range of 1 to 4 mg/ml of what? It is not clear? A polyphenolic extract? How the authors measured these concentrations? |
Thank you for the valuable comment. The methodology lacked this explanation. We apologize for our mistake. Green coffee extracts concentrated to a constant weight (obtained according to "Preparation of green coffee extracts for antioxidant analysis") were made into solutions with concentrations of 1-4mg / ml. The weighed concentrated green coffee extracts were dissolved in the appropriate amount of ethanol (70%) - a solvent commonly used for ABTS, DPPH methods, or in distilled water for FRAP and CUPRAC methods. |
|
What about statistical analysis? |
The statistical analysis methodology has been added in the manuscript. |
|
The missing point of the article is comparing the data obtained regarding antioxidant activity. At least the classical methods (DPPH, FRAP) there are a lot of data’s. |
Thank you for your important comment. The manuscript was supplemented by comparing the antioxidant activity of green coffee extracts with the activity of green tea extract. |
|
These voltametric methods are used for the first time on coffee beans? |
Electrochemical analysis was used to determine the electrochemical behavior of both some chlorogenic acid and green coffee extracts. In our publication, electrochemical methods were used to determine the antioxidant activity and the antioxidant composite index (AC). However, so far electrochemical analyzes and spectrophotometric analyzes have not been compared. Such comprehensive studies have not yet been presented in the literature. |
Reviewer 2 Report
The authors presented a paper analysing green coffee beans for antioxidants. They compared six differing assays, two methods employing electrochemistry, two decolorisation assays - ABTS, DPPH and two redox colorimetric methods - FRAP and CUPRAC. The aim of the paper was to compare these methods and subsequently assess the antioxidant importance of coffee. However, the paper lacked novelty. It is well known that coffee contains antioxidants, and not surprisingly each method yielded positive indications of antioxidants in the coffee extract. All comparitive data was based on the bulk antioxidant properties of the coffee extracts and hence there was no information relating to the selectivity of the testing protocols. More work is required before this paper would be of benefit to the targeted readers.
Author Response
Institute of Polymer and Dye Technology
Technical University of Lodz
90-924 Lodz, ul Stefanowskiego 12/16, Poland
Tel.: +48 42 631 32 23, Fax: +48 42 636 25 43
April 27, 2020
Forests, MDPI
Dear Editor,
We are resubmitting our revised paper entitled “Antioxidant properties of green coffee extract” by, Anna Masek, Malgorzata Latos-Brozio, Joanna Kałużna-Czaplińska, Angelina Rosiak, Ewa Chrzescijanska with a request to reconsider it for publication in Forests (MDPI).
We have carefully considered the Editor and Reviewers' comments. The manuscript was revised exactly according to these comments. The list of responses to the reviewer’s comments and corrections made in the manuscript is attached below.
The manuscript has not been previously published, is not currently submitted for review to any other journal, and will not be submitted elsewhere before a decision is made by this journal.
For correspondence please use the following information:
corresponding author: Anna Masek
Institute of Polymer and Dye Technology
Technical University of Lodz
90-924 Lodz, ul Stefanowskiego 12/16, Poland
Tel.: +48 42 631 32 93
Fax: +48 42 636 25 43
e-mail: anna.masek@p.lodz.pl
Yours sincerely,
PhD, DSc, Anna Masek
Associate Professor
Major Revision
Response to Reviewers:
|
Comments by Reviewer #2 |
Changes by Authors |
|
The authors presented a paper analysing green coffee beans for antioxidants. They compared six differing assays, two methods employing electrochemistry, two decolorisation assays - ABTS, DPPH and two redox colorimetric methods - FRAP and CUPRAC. The aim of the paper was to compare these methods and subsequently assess the antioxidant importance of coffee. However, the paper lacked novelty. It is well known that coffee contains antioxidants, and not surprisingly each method yielded positive indications of antioxidants in the coffee extract. All comparitive data was based on the bulk antioxidant properties of the coffee extracts and hence there was no information relating to the selectivity of the testing protocols. More work is required before this paper would be of benefit to the targeted readers. |
We thank the Reviewer for an important comment. Our mistake, for which we apologize, is the lack of indication of scientific novelty in our manuscript. This mistake has been corrected. The scientific novelty of the work is a combination of different methods for determining the activity of green coffee extracts. So far, no such comprehensive research on green coffee extracts has been done. What's more, spectrophotometric determinations summarize the results for extracts obtained in ethanol and in water, received at different times of brewing coffee. The available literature lacks assessment of the relationship between the type of solvent and antioxidant capacity and reduction of transition metal ions. This publication supplements the deficiencies in the literature. |
Reviewer 3 Report
Dear Authors, thank you for your work, here are some suggestions for improvement:
Method section has considerable amount of flaws, please use full sentences to describe the methods you used and references connected with them (for example DPPH and ABTS). In Gas chromatography section you haven't you should put more effort and make a full sentences. Please describe which instrument, collumn etc., what database have you used, for identification of the chemical compounds. Don’t forget to put the references.
Line 54: You have mentioned HPLC and there Is nothing about it in the rest of the paper.
Line 207: Solutions with concentrations from 1 to 4 mg/ml were tested and it was found that, with increasing concentration, the % inhibition of ABTS and DPPH increased. Please be precise here, what compounds? Be more clear when communicating your results. Connect them with some previous research, there are many good references.
Line 229: Heneczkowski and co-authors replace with Heneczkowski et all.
Line 290: Solutions with concentrations from 1 to 4 mg/ml were tested and it was found that, with increasing concentration, the % inhibition of ABTS and DPPH increased. This is really common knowledge in chemistry. If you increase the concentration of the antioxidants that the antioxidant activity will increase.
Line 308 please delete as you don't have any proof on how it could benefit human health.
I suggest you to be more precise when communicating your results. It was bit confusing to read. Method section need more improvement. It seems like a draft you still need to work on.
I hope my comments were helpful.
Greetings
Author Response
Institute of Polymer and Dye Technology
Technical University of Lodz
90-924 Lodz, ul Stefanowskiego 12/16, Poland
Tel.: +48 42 631 32 23, Fax: +48 42 636 25 43
April 27, 2020
Forests, MDPI
Dear Editor,
We are resubmitting our revised paper entitled “Antioxidant properties of green coffee extract” by, Anna Masek, Malgorzata Latos-Brozio, Joanna Kałużna-Czaplińska, Angelina Rosiak, Ewa Chrzescijanska with a request to reconsider it for publication in Forests (MDPI).
We have carefully considered the Editor and Reviewers' comments. The manuscript was revised exactly according to these comments. The list of responses to the reviewer’s comments and corrections made in the manuscript is attached below.
The manuscript has not been previously published, is not currently submitted for review to any other journal, and will not be submitted elsewhere before a decision is made by this journal.
For correspondence please use the following information:
corresponding author: Anna Masek
Institute of Polymer and Dye Technology
Technical University of Lodz
90-924 Lodz, ul Stefanowskiego 12/16, Poland
Tel.: +48 42 631 32 93
Fax: +48 42 636 25 43
e-mail: anna.masek@p.lodz.pl
Yours sincerely,
PhD, DSc, Anna Masek
Associate Professor
|
Comments by Reviewer #3 |
Changes by Authors |
|
Method section has considerable amount of flaws, please use full sentences to describe the methods you used and references connected with them (for example DPPH and ABTS). |
The methods section has been improved. |
|
In Gas chromatography section you haven't you should put more effort and make a full sentences. Please describe which instrument, collumn etc., what database have you used, for identification of the chemical compounds. Don’t forget to put the references. |
Text was corrected according to the reviewer comments (lines 119-135).
|
|
Line 54: You have mentioned HPLC and there Is nothing about it in the rest of the paper. |
It has been corrected. |
|
Line 207: Solutions with concentrations from 1 to 4 mg/ml were tested and it was found that, with increasing concentration, the % inhibition of ABTS and DPPH increased. Please be precise here, what compounds? Be more clear when communicating your results. Connect them with some previous research, there are many good references. |
The sentence has been corrected. Green coffee extracts prepared in ethanol and water were tested. The test results were compared to the activity of green tea extract. |
|
Line 229: Heneczkowski and co-authors replace with Heneczkowski et all. |
It has been corrected. |
|
Line 290: Solutions with concentrations from 1 to 4 mg/ml were tested and it was found that, with increasing concentration, the % inhibition of ABTS and DPPH increased. This is really common knowledge in chemistry. If you increase the concentration of the antioxidants that the antioxidant activity will increase. |
Thank you for the comment. We agree with the Reviewer. |
|
Line 308 please delete as you don't have any proof on how it could benefit human health. |
It has been corrected. |
|
I suggest you to be more precise when communicating your results. It was bit confusing to read. Method section need more improvement. It seems like a draft you still need to work on. |
Thank you for the valuable comment. The entire manuscript has been carefully examined and corrected. The methods section has been improved. |
Round 2
Reviewer 1 Report
The authors response/modified all the requests.
It can be published in this form.